# Sidewall Corrugation-Modulated Phase-Apodized Silicon Grating Filter

**DOI:** 10.3390/mi15060666

**Published:** 2024-05-21

**Authors:** Wei Jiang, Jijun Feng, Shuo Yuan, Haipeng Liu, Zhiheng Yu, Cunliang Yang, Wenbo Ren, Xincheng Xia, Zhengjie Wang, Fengli Huang

**Affiliations:** 1Shanghai Key Laboratory of Modern Optical System, Engineering Research Center of Optical Instrument and System (Ministry of Education), School of Optical-Electrical and Computer Engineering, University of Shanghai for Science and Technology, Shanghai 200093, China; 2Key Laboratory of Medical Electronics and Digital Health of Zhejiang Province, Jiaxing University, Jiaxing 314001, China

**Keywords:** apodized grating, sidewall corrugation, grating filter, silicon photonics

## Abstract

In this work, phase-apodized silicon grating filters with varying sidewall corrugation width and location were investigated, while the resonance wavelength, extinction ratio, and rejection bandwidth were tuned flexibly. The grating filters with a waveguide width of 500 nm and grating period of 400 nm were fabricated and characterized as a proof of concept. The resonance wavelength of the device can be shifted by 4.54 nm by varying the sidewall corrugation width from 150 to 250 nm. The corresponding rejection bandwidth can be changed from 1.19 to 2.03 nm by applying a sidewall corrugation location offset from 50 to 200 nm. The experimental performances coincide well with the simulation results. The presented sidewall corrugation-modulated apodized grating can be expected to have great application prospects for optical communications and semiconductor lasers.

## 1. Introduction

The silicon photonic integrated circuit has aroused much attention due to its complementary metal–oxide–semiconductor (CMOS) compatibility and strong optical field confinement [1,2,3]. Waveguide grating is more beneficial to realize on-chip integrated optics, which can reflect a specific wavelength to its input port and be used as a band-stop filter in forward transmission and a band-pass filter at the reflecting port. It has been widely used in optical filters [4], high-speed modulators [5], semiconductor lasers [6], optical sensing [7], wavelength-division-multiplexing devices [8], and so on. The subwavelength grating waveguides can be also regarded as types of metamaterials with an engineerable index, mode, and dispersion response [9]. The corresponding spectrum can be designed by specific grating structures [10,11], which can be divided into strip waveguides with symmetrical or asymmetrical sidewall corrugations in the horizontal direction [12,13]. Refractive index modulation brought by an asymmetric grating structure can be used to realize more spectrum adjustment [14,15]. 

For the traditional symmetrical grating, the rejection bandwidth can only be tuned by changing grating length [16], with the resonance wavelength depending on the grating period and duty ratio, and also having to make a trade-off between insertion loss and rejection bandwidth [17,18]. And the coupling coefficient of grating can also be changed by changing the amplitude of grating corrugation [19]. Due to the high refractive index contrast of the SOI platform, the resonance property is strongly influenced by the grating period, duty cycle, and corrugation depth. Asymmetrical grating-based apodization technology can be used to shift the location of grating corrugation, in order to compress the rejection bandwidth [20,21]. Compared with some sub-wavelength gratings with advanced designer functions for spectral filter and mode conversion, the width of the corrugation can be changed to check its influence on the resonance peak [22,23,24]. Recently, multimode waveguide-based apodized grating has aroused much attention [25], which could tune the rejection bandwidth by increasing the corrugation location offset. Here, two TE modes can be excited and modulated simultaneously, inducing a phenomenon contrary to the single-mode waveguide case, while the location offset would counteract the resonance effect and a π-phase offset would usually make the reflection peak disappear [26,27,28]. In order to fully understand the modulation characteristics of the multimode waveguide-based grating, we investigate the influence of the asymmetrical corrugation width and location offset on the grating performance, which can result in a fine tuning of the resonance condition and provide more design flexibility for the grating filters. 

In the following, a sidewall corrugation-modulated phase-apodized silicon grating filter is presented, and the influence of grating corrugation width and location is studied in detail. Due to this, the effective refractive index and coupling coefficient of the grating can be tuned correspondingly, and the rejection bandwidth and resonance wavelength can be changed flexibly. The gratings with a waveguide width of 500 nm and period of 400 nm are fabricated and characterized for a proof of concept, which confirms the tuning property with modulated sidewall corrugations.

## 2. Device Design and Principle

A schematic diagram of the presented grating is shown in Figure 1, which is a strip-shaped silicon waveguide with corrugations on both sidewalls and can be divided into two regions. Figure 1b shows the magnified view with misaligned sidewall corrugations. The width (*W*) and height of the strip waveguide are 500 and 220 nm, respectively. The length (*L*) of the whole grating region is 300 μm with a period (*Λ*) of 400 nm. *P*_1_ and *P*_2_ are the width of sidewall corrugation with a modulation depth (Δ*W*) of 25 nm, and Δ*P* is the location offset of the sidewall corrugations. The increase in sidewall corrugation depth will usually result in a larger coupling coefficient and higher extinction ratio of the resonance peak, as well as a higher transmission loss [29]. Grating geometric parameters are summarized in Table 1.

The working principle of the grating can be explained by the coupled mode theory [30]. For the multi-mode waveguide, if the excited TE_0_ mode in the strip waveguide meets the phase-matching condition, the grating will reversely couple it to the backward TE_1_ mode. On the other hand, if the wavelength does not meet the phase-matching condition, the TE_0_ mode remains intact when propagating through the grating [31]. TE_0_ and TE_1_ modes can be coupled when the phase-matching condition is satisfied, which can be expressed as [32]
(1)$$n_{eff0}+n_{eff1}=\frac{\lambda_{B}}{\Lambda },$$
where *n_eff_*_0_ and *n_eff_*_1_ are the effective refractive indices of TE_0_ and TE_1_ modes, respectively, and *λ*_B_ is the resonance wavelength. Figure 2a shows the electric field distribution of the grating when light propagates from left to right at a resonance wavelength of 1555.91 nm, while Figure 2b is the case at a non-resonance wavelength of 1560 nm. Figure 2c,d are the TE_0_ and TE_1_ mode field distributions of a 500 nm wide straight waveguide, respectively.

The coupling coefficient (*κ*) is usually used to express the resonance effect, which is the magnitude of coupling between the TE_0_ and TE_1_ modes here. For the multimode waveguide-based sidewall-modulated grating, the corrugation location will affect its coupling efficiency, and *κ* can be calculated by [33]
(2)$$\kappa =\left|\frac{\kappa_{0}}{2}+\frac{\kappa_{0}}{2}exp\left(i\cdot 2\pi \Delta P/\Lambda \right)\right|{=\kappa }_{0}sin\left(\frac{\pi \Delta P}{\Lambda }\right),$$
where *κ*_0_ is the maximum grating coupling coefficient, with Δ*P* = *Λ*/2. Here, 2*π*Δ*P*/*Λ* is the phase shift between two grating regions. 

The rejection bandwidth of grating (Δ*λ*) can be determined by the *κ*, which can be approximately calculated by [34]
(3)$$\Delta \lambda =\frac{{\lambda_{B}}^{2}}{n_{group}\times L}\sqrt{1+{\left(\frac{\kappa \times L}{\pi }\right)}^{2}},$$
where *n*_group_ is the average value of the group refractive index of the waveguide grating. 

The influence of grating corrugation width on the spectrum can be understood by the effective refractive index. Considering the grating as a strip waveguide disturbed by sidewall rectangular corrugations, the effective refractive index of the regions with and without corrugations is different. When the sidewall corrugations are asymmetric (*P*_1_ ≠ *P*_2_), the average effective refractive index can be calculated by [35]
(4)$$n_{eff}=\frac{A}{\Lambda }n_{\mathrm{eff}A}+\frac{B}{\Lambda }n_{\mathrm{eff}B}+\frac{C}{\Lambda }n_{\mathrm{eff}C},$$
where *n_eff_*_A_, *n_eff_*_B_, and *n_eff_*_C_ are the effective refractive indices of strip waveguide with double, single, and no corrugation, respectively. A, B, and C are the length of these regions with different waveguide widths in one grating period, as shown in Figure 1b. The increase in waveguide width will make the effective refractive index larger. Because the above three regions have different waveguide widths, which will affect the effective refractive index of each region, there will be different *n_eff_*_A_, *n_eff_*_B_, and *n_eff_*_C_ values, and the effective refractive index is the largest when the strip waveguide has two corrugations. Therefore, when calculating the overall effective refractive index of the grating period, it is necessary to consider the proportion of these areas in the grating period.

The grating was then simulated by the Finite-Difference Time-Domain (FDTD) method using Lumerical FDTD Solutions version 8.24.2387, according to the parameters summarized in Table 1. The perfect matching layer (PML) was adopted for the boundary condition with a grid size of 5 nm. *P*_2_ was varied as 150, 200, and 250 nm with *P*_1_ = 200 nm. At the same time, Δ*P* was varied from 0 to 200 nm to observe its influence on the extinction ratio and rejection bandwidth. Simulated reflection and transmission spectra of the gratings are shown in Figure 3. Figure 3a shows the spectra of grating with *P*_1_ = 200 nm, *P*_2_ = 200 nm, and Δ*P* =200 nm. Figure 3b shows the transmission spectrum when *P*_1_ and *P*_2_ are both 200 nm, with only Δ*P* changed for comparison. With a maximum Δ*P* of 200 nm, the simulated rejection bandwidth is 2.76 nm with an extinction ratio of about 15.79 dB. For multimode gratings, both TE_0_ and TE_1_ modes can be excited but with opposite phases if the phase matching condition is met, so destructive interference will occur when the corrugations with Δ*P* = 0 are aligned. And there is no resonance peak in this case. When Δ*P* decreases, the rejection bandwidth becomes narrower. And the rejection bandwidth is 1.65 nm with a 3.61 dB extinction ratio when the corrugation location offset is 50 nm. Figure 3c shows the transmission spectra when *P*_2_ is varied as 150, 200, and 250 nm with Δ*P* = 200 nm and *P*_1_ = 200 nm. The corresponding resonance wavelengths are 1553.36, 1555.91, and 1557.82 nm, respectively, since the average effective refractive index increases with *P*_2_. The above simulation results show that the resonance wavelength, extinction ratio, and rejection bandwidth can be tuned with both the width and location of the sidewall corrugations.

## 3. Device Fabrication and Characterization

As a proof of concept, the grating chip was fabricated by a standard commercial CMOS foundry for silicon photonics. After ultra-violet (UV) lithography and reactive ion etching (RIE), the SOI wafer was cleaned and deposited with a 3 μm thick SiO_2_ cladding layer. Then, the wafer was cut into different chips for performance characterization. Each grating was fabricated in a U-shaped structure with an input and output straight waveguide spaced by 127 nm for better sidewall coupling with a lensed fiber array, with the microscope image shown in Figure 4b. A U-shaped waveguide without a grating structure is also manufactured, and the transmission and incident power of grating are normalized on this basis. Figure 5 presents the scanning electron microscope (SEM) images of the fabricated gratings. The SEM images are not so clear since the chip was fabricated by commercial foundry, and the silicon dioxide coating had to be removed for scanning electron microscopy measurement by us. The remaining cladding layer would affect the image quality. From the images, the etching may be not so uniform, which may influence the loss and performance of the grating waveguide. Figure 5a–c show the grating with *P*_1_ = 200 nm, and *P*_2_= 200 nm with Δ*P* varying as 50, 100, and 150 nm. Figure 5d–f show the grating with *P*_1_ = 200 nm and Δ*P* = 200 nm but *P*_2_ varying as 150, 200, and 250 nm, respectively.

A schematic diagram of the experimental setup for the grating characterization is shown in Figure 4a, with the inset for the microscope image of the chip under test. A tunable laser was used as the light source, which was connected to a polarization controller and a circulator, and then coupled into the waveguide through a lensed fiber array [36]. The coupling between the fiber and chip is monitored by a charge-coupled device in real-time. The transmitted light was divided into two beams by a 10:90 coupler, while 10% part was used to monitor the coupling status between the fiber array and grating chip, with the other 90% part for the spectrum measurement carried out by an optical spectrum analyzer (OSA). The reflected signal was separated from the incident light path through a circulator and connected to the OSA for reflection spectrum monitoring. Finally, the transmitted light and the reflected light were, respectively, input into the optical spectrum analyzer (OSA) through the optical switch to record the reflection and transmission spectra.

The recorded normalized spectra of the gratings are shown in Figure 6. Figure 6a shows the transmission and reflection peak of grating with *P*_1_ = 200 nm, *P*_2_ = 200 nm, and Δ*P* = 200 nm. Figure 6b shows the corrugation location offset varied transmission spectra with *P*_1_ and *P*_2_ fixed at 200 nm. When Δ*P* = 200 nm (*Λ*/2), a maximum transmission rejection bandwidth of 2.1 nm can be obtained with an extinction ratio of about 8.79 dB. And the rejection bandwidth becomes narrower when Δ*P* gradually decreases, e.g., 1.19 nm with Δ*P* = 50 nm, which confirms that varying the corrugation location offset can effectively tune the rejection bandwidth and extinction ratio. For gratings with *P*_2_ = 150 and 250 nm with a maximum corrugation location offset of 200 nm, the rejection bandwidths are 1.98 and 1.94 nm with extinction ratios of 7.86 and 7.82 dB, respectively, as shown in Figure 6c. For gratings with *P*_2_ = 150, 200, and 250 nm, the measured resonance wavelengths are 1514.24, 1517.05, and 1518.78 nm, respectively. This shows that the resonance wavelength can be tuned by varying the corrugation width by slightly changing the rejection bandwidth and extinction ratio. Thus, more design flexibility for the gratings can be obtained with the modulation of both the sidewall corrugation location and shape. Performances of different Bragg grating filters are compared in Table 2. Compared with other structures, the presented device can realize a fine tuning of the resonance wavelength, rejection bandwidth, and extinction ratio, which can provide more design freedom for the development of grating filters.

It should also be noted that there are still some differences between the experimental results and the simulation spectra, such as the resonance wavelength. This may be caused by the sidewall corrugation shape, which is somewhat a sinusoidal profile as shown in Figure 5. The grating with sinusoidal-shaped corrugations is further simulated for comparison. Figure 7a shows the corresponding electric field distribution of the grating at a resonance wavelength of 1523.03 nm with a corrugation profile, as shown in Figure 5a. And the resonance wavelength has a blue shift, as shown in Figure 7b. Nevertheless, the overall morphology and trend of the experimental spectra can coincide well with the simulation results. The presented method can provide a further understanding of the working principle of the multimode waveguide-based grating.

Furthermore, the sidewall corrugation width and relative location change can provide a fine tuning ability for the resonance condition. For example, the resonance wavelength can be changed by the period and duty ratio of a traditional grating filter. A 10 nm grating period change will result in a wavelength shift of about 28.48 nm for a traditional silicon grating waveguide with period of 400 nm, duty cycle of 0.5, and waveguide width of 450 nm. The fabrication tolerance may be very challenging. But for the sidewall corrugation width change as presented, a 50 nm change will induce a wavelength shift of 1.9 nm, while the extinction ratio can also be tuned by varying the corrugation location offset. The presented fine tuning ability may be helpful for the development of grating spectral filters.

## 4. Conclusions

To summarize, sidewall corrugation-modulated phase-apodized silicon grating filters with a waveguide width of 500 nm and grating period of 400 nm have been demonstrated. The influence of grating sidewall corrugation on the grating spectrum is further studied, which is a new method for changing the grating structure modulation spectrum. The rejection bandwidth can be changed from 1.19 to 2.03 nm with the extinction ratio changing from 1.72 to 8.79 dB by applying a sidewall corrugation location offset from 50 to 200 nm. Furthermore, the resonance wavelength can be shifted by 4.54 nm by varying the sidewall corrugation width from 150 to 250 nm. The proposed sidewall corrugation modulated grating can provide more design freedom for integrated photonic chips such as filters and semiconductor lasers.

## Figures and Tables

**Figure 1 micromachines-15-00666-f001:**
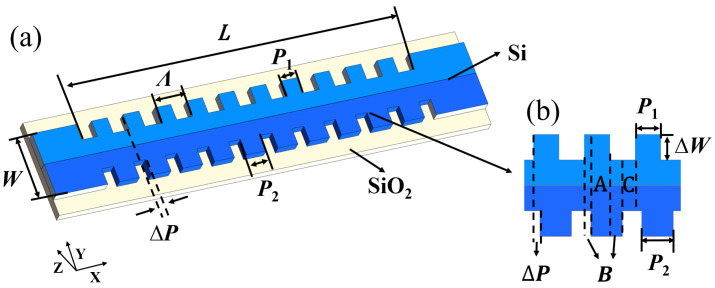
(**a**) Schematic diagram of the grating with (**b**) for the magnified view.

**Figure 2 micromachines-15-00666-f002:**
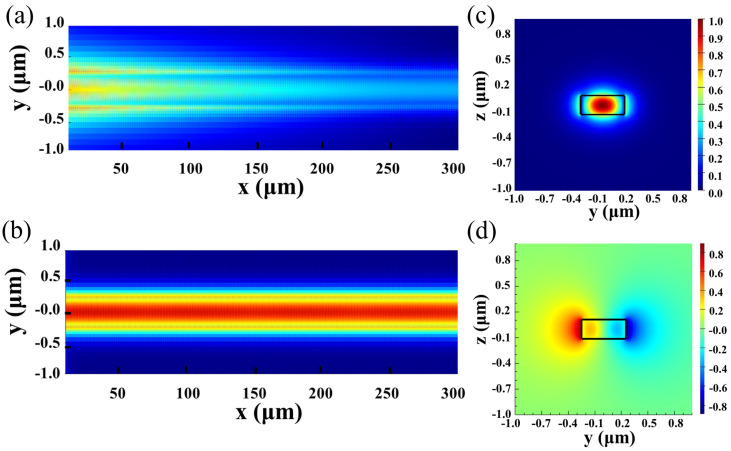
(**a**) Electric field distribution of the grating with light incident from left at a resonance wavelength of 1555.91 nm; (**b**) electric field distribution of grating under non-resonance wavelength (*P*_1_ = 200 nm, *P*_2_ = 200 nm, and Δ*P* = 200 nm) with (**c**,**d**) for the mode field distributions of TE_0_ and TE_1_, respectively.

**Figure 3 micromachines-15-00666-f003:**
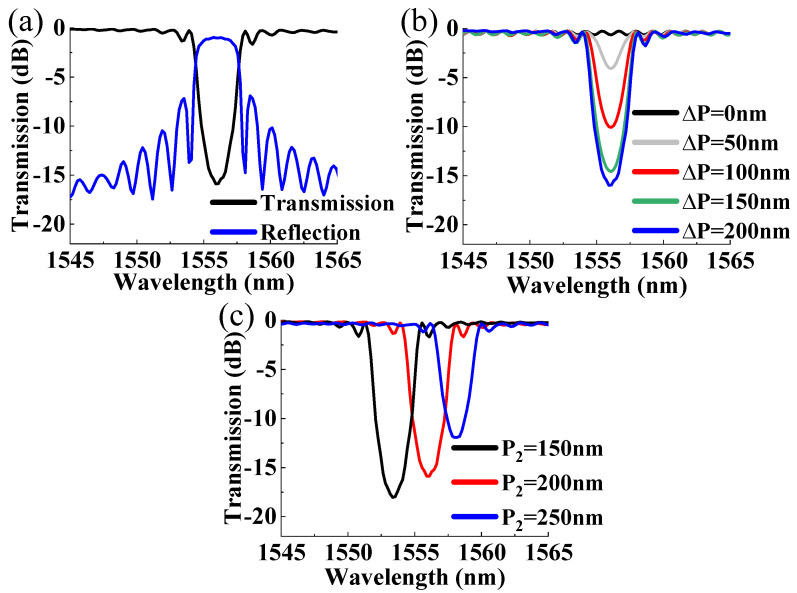
Simulated spectra for grating with (**a**) *P*_1_ = 200 nm, *P*_2_ = 200 nm, and Δ*P* =200 nm; (**b**) varied Δ*P* (*P*_1_ = 200 nm, *P*_2_ = 200 nm); and (**c**) different *P*_2_ (*P*_1_ = 200 nm, Δ*P* = 200 nm).

**Figure 4 micromachines-15-00666-f004:**
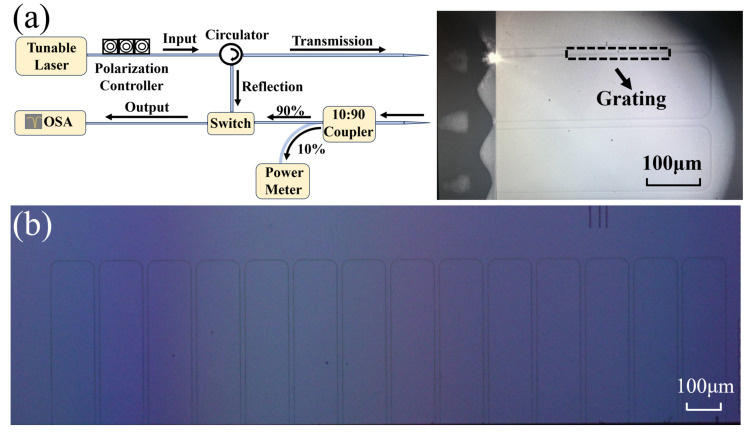
(**a**) Schematic illustration of the experimental setup for the grating chip characterization with (**b**) for the microscope image of the fabricated chip.

**Figure 5 micromachines-15-00666-f005:**
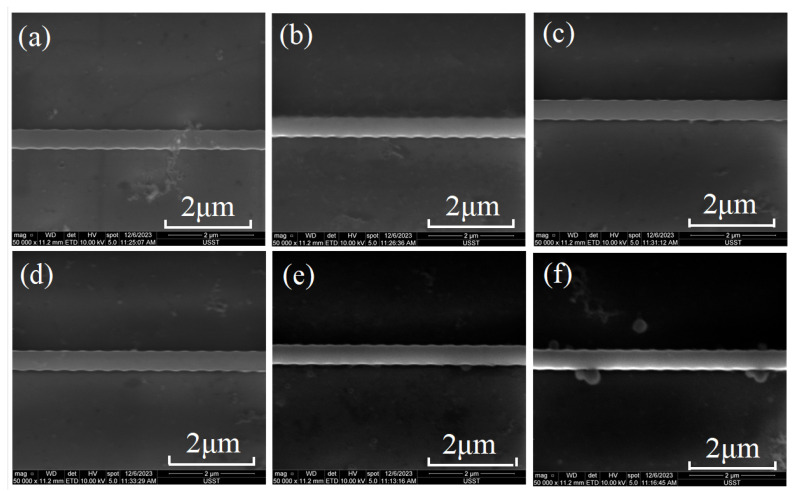
SEM images of gratings with different Δ*P* of (**a**) 50 nm, (**b**) 100 nm, (**c**) 150 nm, and (**d**) 200 nm (*P*_1_ = 200 nm, *P*_2_ = 200 nm), with (**e**,**f**) for varying corrugation width *P*_2_ of 150 and 250 nm, respectively (*P*_1_ = 200 nm, Δ*P* = 200 nm).

**Figure 6 micromachines-15-00666-f006:**
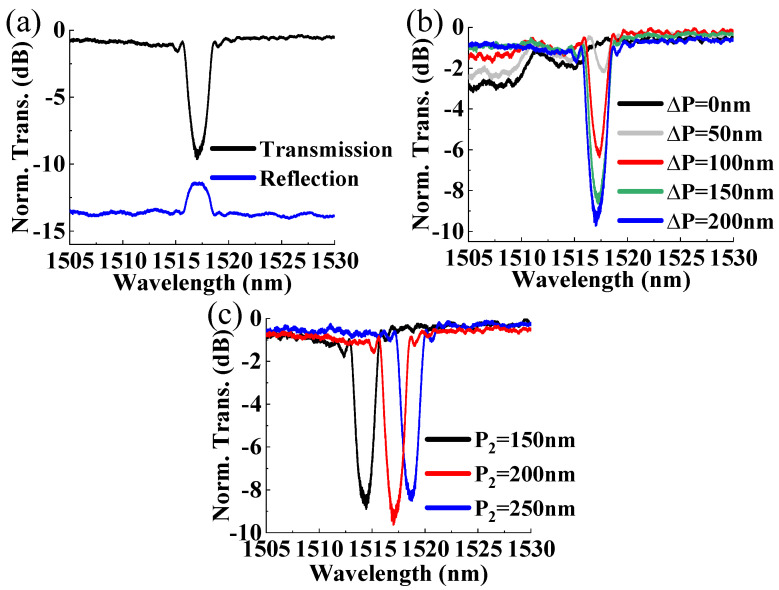
Measured spectra for (**a**) with *P*_1_ = 200 nm, *P*_2_ = 200 nm, and Δ*P* = 200 nm, (**b**) with varied Δ*P* (*P*_1_ = 200 nm, *P*_2_ = 200 nm), and (**c**) with varied *P*_2_ (*P*_1_ = 200 nm, Δ*P* = 200 nm).

**Figure 7 micromachines-15-00666-f007:**
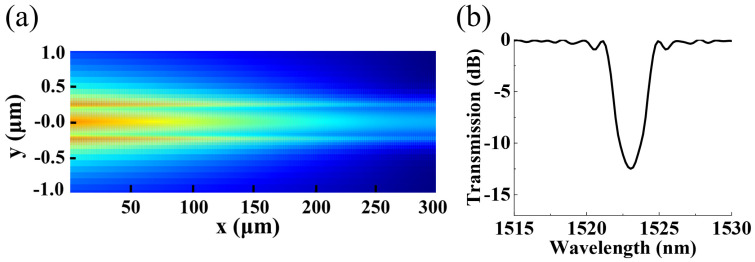
(**a**) Electric field distribution of the sinusoidal-shaped grating for light incident from left to right at a resonance wavelength of 1523.03 nm, with (**b**) for the corresponding simulated transmission spectra.

**Table 1 micromachines-15-00666-t001:** Geometric parameters of the presented silicon grating filter.

Grating Parameters	Geometric Parameters
Waveguide width (*W*)	500 nm
Waveguide height	220 nm
Period (*Λ*)	400 nm
Corrugations etching depth (Δ*W*)	25 nm
Grating length (*L*)	300 μm
Sidewall ripple width (*P*_1_)	200 nm
Sidewall ripple width (*P*_2_)	150, 200, 250 nm
Sidewall corrugation location offset (Δ*P*)	0, 50, 100, 150, 200 nm

**Table 2 micromachines-15-00666-t002:** Performance comparison between different Bragg gratings.

	Rejection Bandwidth	Resonance Wavelength	Extinction Ratio	Fine Tuning	Footprint
Tradition grating [16]	5 nm	1493 nm	15 dB	No	620 × 0.5 μm^2^
Misaligned corrugation grating [14]	~20 nm	1547 nm	28 dB	No	92 × 0.55 μm^2^
Cladding-Modulated Bragg Grating [15]	~50 nm	1523 nm	7.98 dB	No	16.4 × 3 μm^2^
π-phase shift grating [28]	~4 nm	1620 nm	15 dB	No	100 × 0.6 μm^2^
This work	2.03 nm	1517.05 nm	8.79 dB	Yes	300 × 0.5 μm^2^

## Data Availability

The original contributions presented in the study are included in the article, further inquiries can be directed to the corresponding authors.

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
