# Peer review of "Sidewall Corrugation-Modulated Phase-Apodized Silicon Grating Filter"

_micromachines, 2024, doi:10.3390/mi15060666_

Round 1
Reviewer 1 Report
Comments and Suggestions for Authors
I have the following suggestions:
1) I am confused why there is no Bragg reflection in the transmission spectrum when deltaP=0 in figure 3 (b)?
2) I am not sure if the author has misused the word "ridges". In literature, it has been used as "side wall corrugations".
3) Figure 2 (a) should be plotted again for Bragg reflection state and non-Bragg reflection state.
4) What is the influence of the side wall corrugation depth (deltaW) on the transmission spectrum?
5) From figure 3 (b), it can be seen that ER was improving with increasing deltaP, then why author stopped at 200 nm?
6) Why SEM images are not clear?
7) What is the role of the power meter (10:90) when the reflection spectrum is monitored via OSA?
8) Provide a table and compare the performance of BG demonstrated in previous works with the work presented by the authors, such as: https://www.mdpi.com/2304-6732/11/2/158; https://photonics.pl/PLP/index.php/letters/article/view/12-28; https://opg.optica.org/ol/abstract.cfm?uri=ol-42-15-3040, and many more.
9) Moreover a table is required in the device model section which defines all the geometric parameters and the meaning of their abbreviations that are used in the device design.
10) The detail on simulation software and parameters should be added.
11) The fabrication process is also not presented in detail such as detail on Litho, etching etc.
Comments on the Quality of English LanguageNone.
Reviewer 2 Report
Comments and Suggestions for Authors
In this work, the authors have a ridge waveguide with sidewalls asymmetric gratings, which can act as an optical filter. They try to vary the width and relative positions of the gratings corrugations to tune the filter center-wavelength, rejection bandwidth, and extinction ratio. That is of course a well-known effect that has been published in several previous literatures, for example:
- http://dx.doi.org/10.1364/OL.41.002450
- https://doi.org/10.1364/OE.20.015547
- https://doi.org/10.1109/LPT.2010.2103305
- https://doi.org/10.1007/s12200-018-0813-1
- http://dx.doi.org/10.1364/OL.41.005039
- http://dx.doi.org/10.1364/OL.39.005519
- https://doi.org/10.1364/OL.43.003144
- http://dx.doi.org/10.1364/OE.24.026901
The filter achieved tuning results that are not significant enough for publications. The change of center wavelength by just 4.54nm, rejection bandwidth by only 0.84nm, or achieving a maximum extinction ratio of just 8.79dB are all small values that are considered non-significant.
Additional comments:
- - The authors actually do not change the corrugations width and their offset position by any active means like electrical or thermal methods. They just change it during fabrication. Which cannot be considered tuning in the first place.
- - The multimode operation and role are not clearly explained in the manuscript.
- - The quality of Figure 4 is very bad.
- - The experimental multi-modes profiles of fabricated waveguide are missing in the manuscript.
Reviewer 3 Report
Comments and Suggestions for Authors
In the manuscript, the authors reported apodized grating filters for silicon (Si) waveguides for wavelength filtering. Numerical simulations and theoretical analysis on tuning grating parameters are given, with in general agreement with experimental results.
Before publication, the following technical comments should be properly addressed.
1. Apodized gratings can indeed provide more degrees of freedom on structure tuning if compared to conventional gratings. Nevertheless, the subwavelength grating waveguides can be also regarded as types of metamaterials with engineerable index, mode and dispersion response (see Ref: Nature, 560, 565–572, 2018).
Besides conventional gratings, the authors should also mention and compare the proposed device with other subwavelength grating (SWG) waveguides (like Ref DOI: 10.1002/lpor.202300485), or meta-waveguides (Ref: Light: Science & Applications, 10, 235, 2021. Ref: Photonics Research, 8, 564-576, 2020) with advanced designer functions for spectral filter and mode conversion. The abovementioned references should be benchmarked to further highlight the novelty of this work.
2. The device fabrication looks not ideal enough. Accurate implementation of the apodization demand on precise device nanofabrication. The actual device gragting structure and periods also impact device performance. Some questions are given here regarding section 3.
(1) The authors mentioned their devices are patterned by UV photolithography. However, how are the device etched (as I assume it would be based on SOI wafers)? Are they dry etched by ICP RIE or using wet etching method? Some device fab details are not clearly manifested.
(2) For UV photolithography tools at universities, they are typically based on contact exposure mode mask aligners, which has very limited resolution ~ 1 um. To realize the sub-micron fine features proposed by the authors for SWG, usually e-beam lithography (EBL) is required.
Based on the SEM results in Fig. 4, they are still quite a distance from the designs in Fig. 1 with unsatisfactory device fabrication quality. The authors need to comment on their device nanofabrication and why not choosing EBL for devices with much higher fabrication quality and resolution.
(3) The color contrast in Fig. 4 sub-panels around the waveguides also indicate ununiform etching quality, or photoresist residues (panels 4a-4e, and particle/polymer residues for 4g). These also impact waveguide loss and performance that would need the authors comments on this regard.
3. The coupled mode theory used here looks class model reiteration (Eqs. 1~2). The authors can add more comments or practical considerations on how to properly design the parameters such as grating period or the apodization factor ΔP to design filer at certain given wavelength or mode.
4. The U-shaped waveguide designs can be convenient for the measurement setups indicated in Fig. 6. However, this U-waveguide bending (looks with a bending radius < 30 um) will also induce certain loss and reflection that can change the reflected light power if compared to straight waveguides. Did the authors also make comparisons of same U-shaped waveguides with and without gratings to see if this can impact the transmitted or reflected power (i.e. some reflection cased by waveguide bending instead of gratings)?
Round 2
Reviewer 1 Report
Comments and Suggestions for Authors
I am willing to accept the paper in its current form.
Comments on the Quality of English LanguageNone.
Reviewer 2 Report
Comments and Suggestions for Authors
No more comments